# Landscape Analysis of *COL6A1*, *COL6A2*, and *COL6A3* Pathogenic Variants in a Large Italian Cohort Presenting with Collagen VI-Related Myopathies: A Nationwide Report

**DOI:** 10.3390/biom15101426

**Published:** 2025-10-08

**Authors:** Fernanda Fortunato, Laura Fiocco, Alice Margutti, Marcella Neri, Adele D’Amico, Enrico Bertini, Enzo Ricci, Eugenio Maria Mercuri, Marika Pane, Roberto Massa, Giulia Greco, Angela Lucia Berardinelli, Cristina Cereda, Antonella Pini, Luciano Merlini, Carlo Fusco, Carmelo Rodolico, Sonia Messina, Chiara Fiorillo, Claudio Bruno, Marina Pedemonte, Monica Traverso, Isabella Moroni, Lorenzo Maggi, Sara Gibertini, Elena Pegoraro, Esther Picillo, Luisa Politano, Marianna Scutifero, Fabiana Vercellino, Francesca Massaro, Massimiliano Filosto, Paolo Gasparini, Federica Ricci, Tiziana Enrica Mongini, Rita Selvatici, Alessandra Ferlini, Francesca Gualandi

**Affiliations:** 1Medical Genetics Unit, Department of Medical Sciences, University of Ferrara and Department of Mother and Child, University Hospital S. Anna Ferrara, 44121 Ferrara, Italy; frtfnn@unife.it (F.F.); laura.fiocco@unife.it (L.F.); mrglca@unife.it (A.M.); marcella.neri@unife.it (M.N.); mrllcn@unife.it (L.M.); svr@unife.it (R.S.); gdf@unife.it (F.G.); 2Unit of Neuromuscular and Neurodegenerative Disorders, Bambino Gesù Pediatric Hospital, IRCCS, 00146 Rome, Italy; adele2.damico@opbg.net (A.D.); enricosilvio.bertini@opbg.net (E.B.); 3Unit of Neurology, IRCCS A. Gemelli University Polyclinic Foundation, 00168 Rome, Italy; enzo.ricci@policlinicogemelli.it; 4Nemo Clinical Center, IRCCS A. Gemelli University Polyclinic Foundation, 00168 Rome, Italy; eugeniomaria.mercuri@policlinicogemelli.it (E.M.M.); marika.pane@policlinicogemelli.it (M.P.); 5Pediatric Neurology, Catholic University of the Sacred Heart, 00168 Rome, Italy; 6Unit of Neuromuscular Disorders, Department of Systems Medicine, Tor Vergata University of Rome, 00133 Rome, Italy; massa@uniroma2.it (R.M.); gl.grc.9@gmail.com (G.G.); 7Department of Child Neurology and Psychiatry, IRCCS Mondino Foundation, 27100 Pavia, Italy; angela.berardinelli@mondino.it; 8Center of Functional Genomics and Rare Diseases, Buzzi Children’s Hospital, 20154 Milan, Italy; cristina.cereda@asst-fbf-sacco.it; 9Department of Biomedical and Clinical Sciences (DIBIC), University of Milan, 20122 Milan, Italy; 10Pediatric Neuromuscular Unit, IRCCS Institute of Neurological Sciences of Bologna, 40139 Bologna, Italy; antonella.pini@isnb.it; 11Department of Biomedical and Neuromotor Science, DIBINEM, University of Bologna, 40136 Bologna, Italy; 12Child Neurology and Psychiatry Unit, Presidio Ospedaliero Santa Maria Nuova, AUSL-IRCCS di Reggio Emilia, 42123 Reggio Emilia, Italy; carlo.fusco@ausl.re.it; 13Unit of Neurodegenerative Diseases, Department of Clinical and Experimental Medicine, University of Messina, 98122 Messina, Italy; crodolico@unime.it (C.R.); sonia.messina@unime.it (S.M.); 14Child Neuropsichiatry Unit, IRCCS Istituto Giannina Gaslini, 16147 Genoa, Italy; chiara.fiorillo@edu.unige.it; 15Department of Neurosciences, Rehabilitation, Ophthalmology, Genetics, Maternal and Child Health (DINOGMI), University of Genoa, 16132 Genoa, Italy; claudiobruno@gaslini.org; 16Center of Translational and Experimental Myology, IRCCS Istituto Giannina Gaslini, 16147 Genoa, Italy; 17Pediatric Neurology and Muscle Disease Unit, IRCCS Istituto Giannina Gaslini, 16147 Genova, Italy; marinapedemonte@gaslini.org; 18Medical Genetics Unit, IRCCS Istituto Giannina Gaslini, 16147 Genoa, Italy; monicatraverso@gaslini.org; 19Department of Pediatric Neurosciences, Fondazione IRCCS Istituto Neurologico Carlo Besta, 20133 Milan, Italy; isabella.moroni@istituto-besta.it; 20Neuroimmunology and Neuromuscular Diseases Unit, Fondazione IRCCS Istituto Neurologico Carlo Besta, 20133 Milan, Italy; lorenzo.maggi@istituto-besta.it (L.M.); sara.gibertini@istituto-besta.it (S.G.); 21Department of Neuroscience, University of Padua, 35122 Padua, Italy; elena.pegoraro@unipd.it; 22Cardiomyology and Medical Genetics, University of Campania “Luigi Vanvitelli”, 80138 Napoli, Italy; esther.picillo@gmail.com (E.P.); luisa.politano@unicampania.it (L.P.); marianna.scutifero@policliniconapoli.it (M.S.); 23Child Neuropsychiatry Unit, SS. Antonio e Biagio e Cesare Arrigo Hospital, 15121 Alessandria, Italy; fvercellino@ospedale.al.it; 24Neurology Unit, Santo Stefano Hospital, 59100 Prato, Italy; francesca2.massaro@uslcentro.toscana.it; 25Department of Clinical and Experimental Sciences, University of Brescia, 25123 Brescia, Italy; massimiliano.filosto@unibs.it; 26NeMO-Brescia Clinical Center for Neuromuscular Diseases, ASST Spedali Civili, 25123 Brescia, Italy; 27Institute for Maternal and Child Health-IRCCS, Burlo Garofolo, 34137 Trieste, Italy; paolo.gasparini@burlo.trieste.it; 28Department of Medicine, Surgery and Health Sciences, University of Trieste, 34127 Trieste, Italy; 29Neuromuscular Unit, Department of Neurosciences “Rita Levi Montalcini”, University of Turin, 10126 Turin, Italy; federica.ricci@unito.it (F.R.); tizianaenrica.mongini@unito.it (T.E.M.)

**Keywords:** collagen VI-related myopathy, *COL6A1* gene, *COL6A2* gene, *COL6A3* gene

## Abstract

Collagen VI is an extracellular matrix component encoded by *COL6A1*, *COL6A2* and *COL6A3* genes. Causative variants in these genes are associated with the following collagen VI-related myopathies: severe Ullrich congenital muscular dystrophy (UCMD), milder Bethlem myopathy (BM) and intermediate phenotypes (INT). We report the mutation landscape of *COL6A* genes in 138 Italian patients affected with a collagen VI-related phenotype. The patient cohort included 44 (32%) UCMD, 9 (7%) INT, 61 (44%) BM and 21 (15%) INT/BM patients; 3 patients (2%) with a myosclerosis myopathy (MM) phenotype were also considered. We identified 104 different variants: 26 in *COL6A1* (25%), 52 in *COL6A2* (50%) and 26 in *COL6A3* (25%). The variant spectrum includes missense, splicing, small indel, frameshifting and nonsense variants. Glycine substitutions in the triple helical domain of the collagen VI protein are the commonest variants and occur in all phenotypes. Our genetic profiling disclosed a unique mutation scenario and phenotypic association of the *COL6A2* gene with respect to *COL6A1* and *COL6A3*, which may be related to a different evolutive history. Landscape mutation analysis of variants occurring in ultrarare conditions, such as collagen VI-related myopathies, is crucial to better understand the variations’ profile and to gain insight into fundamental knowledge about gene structure and its evolutive origin.

## 1. Introduction

Collagen VI is an extracellular matrix component encoded by *COL6A1*, *COL6A2* and *COL6A3* genes.

The *COL6A1* (OMIM *120220) and *COL6A2* (OMIM *120240) genes are located on chromosome 21q22.3, whereas *COL6A3* (OMIM *120250) is located on chromosome 2q37. The *COL6A1* gene is subdivided into 35 exons over an area of 23 kb and is transcribed into a single mRNA of approximately 4.2 Kb [1]. The *COL6A2* gene (spanning 28 exons over an area of 35 kb) and the *COL6A3* gene (including 44 exons spanning 100 kb) are transcribed in multiple isoforms, undergoing extensive alternative splicing [2,3].

Collagen VI is composed of three main α-chains, the α1 (VI), α2 (VI)—encoded by *COL6A1* and *COL6A2*, respectively—and α3(VI)—encoded by *COL6A3—*and organized into a network of microfibrils with an essential role in anchoring the basement membrane to the extracellular matrix (ECM) [4].

Each chain contains a central triple helical domain (THD) with repeating Gly-X-Y subunits, flanked by extensive N- and C-terminal domains with homology to the type A domains of von Willebrand factor [5].

Collagen VI is broadly expressed with the highest representation in connective tissues, including bone, skin, tendon, cartilage and interstitial fibroblasts of skeletal muscle; expression in the central and peripheral nervous system, intestine, lung, adipose tissue, pancreatic islets, ovarian follicles, kidney glomeruli, vasculature and cornea has also been reported [6].

Causative variants in *COL6A1*, *COL6A2* and *COL6A3* genes result in skeletal muscle disorders, collectively called “collagen VI myopathies” [7], which represent a common type of congenital muscular dystrophy (CMD), identified in 20.24% of Italian CMD patients [8].

Collagen VI-related myopathies constitute a continuum of overlapping clinical phenotypes with Ullrich congenital muscular dystrophy (UCMD; OMIM #254090) at the more severe end and Bethlem myopathy (BM; OMIM #158810) at the milder end; moreover, intermediate phenotypes of various severities have been described and a new category, intermediate collagen VI-related myopathy, has been suggested for patients with a clinical picture between the two extreme ends [9]. An additional, mostly contractural, phenotype is referred to as myosclerosis myopathy (MM; OMIM #255600) [10].

Several nationwide studies have described genetic mutations in *COL6A* genes. Glycine amino acid substitutions in the THD are the commonest mutations, accounting for about 30% of known pathogenic variations [9,10,11].

Other missense, nonsense, frameshifting and splicing variants, and some rare large genomic deletions have been identified [12,13]. A frequent intronic variant causing pseudoexon recognition and inclusion in *COL6A1* has also been reported [14]. Interestingly, variants did not show ethnic specificity [7,15,16,17].

Dominant and recessive mutations occur in *COL6A1*, *COL6A2* and *COL6A3* and result in a clinical severity depending on the impact of the mutation on the multimeric complex of collagen VI and its assembly process.

Beside inherited variants, more than 50% of UCMD cases harbor monoallelic, de novo dominant mutations (all types), including those affecting the Gly-X-Y subunits in the N-terminal end of THD. These mutations exert dominant negative effects on the assembly and structure of collagen VI, making the mutant chain unable to perform tetrameric assembly [18,19]. In contrast, mutations in rare recessive cases clustering near the C-terminal of THD or in the C-terminal domain disrupt the initial formation of monomers and consequently prevent their inclusion in the assembly process [11].

The present study reports on an Italian nationwide study describing the *COL6A* genes mutation scenario in a cohort of 138 patients affected by collagen VI-related myopathies.

Genetic diagnosis is fundamental to establishing genotype/phenotype relationships, facilitating biomarker discovery and patients’ enrollment in personalized therapies.

## 2. Materials and Methods



**Patient cohort**



We studied 138 patients with clinically and genetically confirmed diagnosis of collagen VI-related myopathies referred by 21 Italian diagnostic centers—13 from North Italy, 5 from Central Italy and 3 from South Italy—in a temporal window of 15 years (2007–2022). The study was conducted within the routine diagnostic flowchart and for diagnostic purposes.

Ethical consent was collected in each center as part of the routine diagnostic procedures for *COL6A* genetic diagnosis and the study was approved by Comitato Etico di Area Vasta Emilia Centro (CE-AVEC) 66/2020/Oss/AOUFe 2020-01-23.

Affected family members of probands were not included in the present study.

The clinical phenotype of patients was defined according to the output of the 166th ENMC International Workshop on Collagen VI Myopathies, where the UCMD phenotype refers to patients who have never walked or have lost the ability to walk by the age of 12, the BM phenotype refers to patients who are able to walk during adulthood (>19 years) and the INT phenotype refers to patients who lose ambulation during their teens (13–19 years) [20]. The myosclerosis myopathy phenotype [10] was also considered, as was an INT/BM phenotype (defined for patients still able to walk at the last clinical evaluation, performed between 12 and 19 years).



**Genetic analysis**



Genetic testing was performed by Sanger sequencing [21] and/or next-generation sequencing (NGS) custom panels. In selected cases, a custom oligonucleotide CGH array was used [22].

NGS analysis of *COL6A* genes was performed through a custom panel called “NEUROMIO”, including 171 neuromuscular genes, of which 80 were for hereditary neuropathies, 46 for congenital myopathies, 42 for muscular dystrophies and 3 for motoneuron iseases.

Library preparation was performed using the Illumina DNA Prep with Enrichment (target enrichment) method before proceeding to paired-end sequencing (150 bp) on the MiSeq™Dx (Illumina Inc., San Diego, CA, USA). The average coverage (Miseq) was 500×.

Coverage achieved from the entire NGS panel was 99.8% at a read depth > 20×; coverage of *COL6A* genes was 100%. Sequence analysis included the coding exons and flanking intronic regions (±50 bp upstream and downstream of each exon). Specific probes were designed to ensure that intron 11 of *COL6A1* was captured.

Standard analysis (alignment, annotation, filtering and prioritization) of the raw data included sequence alignment and variants (e.g., SNP, InDel and copy number variations – CNVs-) calling and annotation were performed using Emedgene v37.0.0 software Illumina (based on the genome assembly Homo sapiens GRCh37-hg19). The integrative genomics viewer [23] was also used to visualize the read depth and the quality of the reads.

Variants with a frequency exceeding 0.1% in population databases, including the Exome Variant Server (ESP), the Exome Aggregation Consortium (ExAC) and gnomAD, were considered common and excluded from further analysis.

All identified variants were classified, according to American College of Medical Genetics and Genomics (ACMG) guidelines, into one of 5 classes: pathogenic (P), likely pathogenic (LP), uncertain significance (VUS), likely benign (LB) and benign (B) [24]. Varsome [25] and/or ClinVar [26] and/or Franklin [27] were used as tools to sum up actual knowledge about the variants. Variants classified as LP or P were considered, as were VUS satisfying the PM2 criterion (“absent from controls”) of the ACMG guidelines. The molecular confirmation of variants was performed by standard Sanger sequencing on an automated analyzer (Applied Biosystems ^®^3130xl and/or 3500DX, Thermo Fisher Scientific Inc. Waltham, MA, USA).

## 3. Results

Clinical and genetic data of *COL6A1*, *COL6A2* and *COL6A3* patients are presented in Table 1.

A summary of VUS characteristics identified in *COL6A1*, *COL6A2* and *COL6A3* genes is reported in Appendix A.

### 3.1. COL6A Genes-Identified Variants

Twenty-six different *COL6A1* variants were identified in 48 patients (35% of the total cohort) (45 heterozygous, 1 compound heterozygous and 2 homozygous), with a predominance of splicing and missense variants.

In the *COL6A2* gene, 52 different variants were detected in 56 patients (40% of the total cohort) (32 heterozygous, 12 compound heterozygous and 12 homozygous), encompassing all mutation types, with a predominance of missense variants.

Twenty-six *COL6A3* variants, mostly missense, were identified in 34 patients (25%) (31 heterozygous, 1 compound heterozygous and 2 homozygous) (Figure 1a,b).

For all detected variants, a pathogenicity prediction analysis was performed according to the ACMG criteria, resulting in the following classification:41 pathogenic variants (P)39 probably pathogenic variants (LP)24 variants of uncertain significance (VUS), satisfying the ACMG PM2 criterion (Table 1 and Appendix A).

### 3.2. COL6A Genes and Phenotypes

Overall, in our study, we identified 44 (32%) UCMD, 9 (7%) INT, 61 (44%) BM and 21 (15%) INT/BM patients. Moreover, three patients (2%) presented with an MM phenotype (Figure 2a).

*COL6A1* variants were associated with all phenotypes except MM (20 UCMD, 3 INT, 8 INT/BM, 17 BM).

The MM phenotype was present only in the *COL6A2* cohort, where less severe phenotypes predominated (15 UCMD, 5 INT, 6 INT/BM, 27 BM, 3 MM).

In the *COL6A3* gene, a similar prevalence of less severe phenotypes was observed (9 UCMD, 1 INT, 7 INT/BM, 17 BM) (Figure 2b).

### 3.3. COL6A Genes, Phenotypes and Inheritance

Most patients (108/138—78.3%) presented a monoallelic *COL6A* gene variant, suggesting an autosomal dominant/de novo inheritance. Multiple (two or more) variants were identified in 30/138 (21.7%) patients, suggesting a recessive inheritance.

Although monoallelic variants were prevalent in all *COL6A* genes, biallelic variants were significantly represented in the *COL6A2* gene; indeed, biallelic variants in the *COL6A2* gene were identified in 24 patients, whereas only three patients carried biallelic variants in *COL6A1* and *COL6A3*.

Biallelic variants were exclusively associated with UCMD in the *COL6A1* cohort, whereas in the *COL6A2* and *COL6A3* cohorts, they were linked to different phenotypes, with BM being the most prevalent in the *COL6A2* cohort (Figure 3).

### 3.4. COL6A Variants and Protein Domain Distribution

In the *COL6A1* and *COL6A3* genes, the identified mutations were preferentially located in the N-terminal and THD domains, whereas they were rare in the C-terminal domain. However, in the *COL6A2* gene, the mutations were distributed throughout the gene, with a prevalence in the C-terminal domain.

The mutation distribution within different domains of the three genes is reported in Figure 4.

In detail, for the α1(VI) chain, THD was the most affected domain, with missense variants (glycine residue substitutions only) and splicing variants being the most common mutations.

For the α2(VI) chain, the variants were equally distributed between the THD and the C-terminal domain; in both regions, missense mutations were the most frequent.

In the α3(VI) chain, variants, mostly missense, were equally distributed in the THD and in the extended N-terminal region.

## 4. Discussion

We report on the largest study describing a nationwide genetic landscape of unrelated patients with collagen VI-related myopathies.

The cohort of 138 patients came from 21 Italian diagnostic centers, distributed throughout the national territory, diagnosed in a temporal window of 15 years (2007–2022).

Mutation detection was based on sequencing, either traditional Sanger sequencing or the next-generation sequencing approach, due to the evolution of technologies over time. Only in a few cases was a custom oligonucleotide CGH array used. This strategy implies that the occurrence of CNVs in *COL6A* genes was not fully investigated. Nevertheless, based on the Human Gene Mutation Database (HGMD), large genomic rearrangements account for a small proportion of collagen VI-related myopathy cases (≈5%) [28].

In our cohort, the proportion of *COL6A1* (35%), *COL6A2* (40%) and *COL6A3* (25%) variants substantially overlapped with those already described in Western countries, reported to be 38%, 44% and 18%, respectively [29]. Similar proportions were found in an Egyptian cohort of 23 patients [17], but not in a larger cohort of 119 subjects with collagen VI-related myopathies of Spanish and American origin studied by Natera-de Benito and colleagues [30] or in recent sets of Asian cases in which *COL6A1* variants were predominant [7,31,32]; however, the analysis of another Chinese cohort of patients showed a preponderance of *COL6A2* variants [16].

This mutational scenario does not seem to support the existence of clear population or ethnic differences in the involvement of *COL6A* genes. Common to all reported studies, *COL6A3* seems to be the least represented in cohorts of patients with collagen VI-related myopathies. Notably, it has been proposed that the α5(VI) and α6(VI) chains, highly homologous with α3(VI), can substitute for α3(VI) in assembling with α1(VI) and α2 (VI), perhaps compensating for the loss of the α3(VI) chain, if mutated.

The identified mutational scenario, including 104 *COL6A* different variants, further highlights the known considerable allelic heterogeneity of collagen VI-related myopathies [33]. According to the HGMD, more than 2000 different pathogenic variants have been reported in the *COL6A* genes, mainly point mutations, which account for ≈82% of variants [28].

In our cohort, the variant spectrum encompassed missense, splicing, frameshift/nonsense variants and in-frame deletions/insertion. Interestingly, the *COL6A2* gene was the only one in which we observed the entire spectrum of variants; this gene also showed the highest allelic variability, with a total of 52 different variants identified.

Missense variants were the most frequent in all genes (52.9% of the total number of variants detected), followed by splicing variants. This finding is consistent with the HGMD data mentioned above and with the results of other studies [7,16,32].

Almost all variants in the *COL6A1* and *COL6A3* genes were detected in heterozygosity (monoallelic), whereas variants in compound heterozygosity/homozygosity were commonly observed in the *COL6A2* gene, associated with both UCMD and BM phenotypes.

Among the missense variants, heterozygous glycine substitutions interrupting the triple Gly-X-Y repetitive sequence in α(VI) chains and well recognized as pathogenic [11,31] accounted for 49% of the total missense variants; among these, the *COL6A1* p.(Gly284Arg) and p.(Gly293Arg) variants recur in our cohort, being detected in eight and six different individuals, respectively and associated with the full phenotypic spectrum.

Phenotypic variability in collagen VI-related myopathies is common and well documented in the literature [34]. It has been suggested that genetic modifiers, epigenetic and/or environmental factors or the occurrence of mosaicism may be involved in the explanation of this variability [35,36,37]. The availability of groups of patients with the same *COL6A* gene genotype but different degrees of severity could represent a useful tool to further investigate mechanisms underlying phenotypic variability.

Splicing variants were the second most common variant type (25.9% of the total number of variants detected), whereas they were the most common variants identified in other works [30]. Our cohort included four female patients carrying the deep intronic variant in *COL6A1* (c.930+189C>T), described as a recurrent mutation that disrupts the Gly-X-Y motif at the N-terminal end of the THD and leads to a dominantly acting in-frame pseudo-exon insertion [14]. All identified patients shared a severe UCMD phenotype and early loss of ambulation, as described in previously reported cases [38]. In fact, previous studies have suggested that this recurrent intronic variant has a more pronounced dominant-negative action than other dominantly acting mutations in the *COL6A* gene, leading to a distinct phenotype of UCMD [38,39].

This novel and unexpectedly common *COL6A1* intronic variant is important to identify, especially considering that it has been shown to be amenable to exon-skipping therapy [39].

Less common in our cohort were nonsense variants (found almost exclusively in the *COL6A2* gene), frameshifts, small deletions or in-frame insertions. Variants introducing a premature stop codon (nonsense, frameshift and some splicing variants) were more common in recessive forms and more often associated with severe phenotypes, according to the literature data [30].

As previously discussed, collagen VI myopathies are caused either by recessive loss-of-function mutations or, more commonly, by de novo dominant-negative pathogenic variants in the three major *COL6A* genes. These dominant-negative variants generally fall into two categories: in-frame deletions/insertions and single nucleotide missense mutations that substitute glycine residues, thereby disrupting the Gly-X-Y motif of the THD. Nevertheless, nonsense or truncating mutations can also exert a dominant-negative effect if nonsense-mediated decay is escaped and the resulting truncated protein fragments are assembled with normal collagen chains, thus altering supramolecular structures [40].

In our cohort, we also identified, through a custom oligonucleotide CGH array, a deletion within intron 1A of the *COL6A2* gene occurring in compound heterozygosity with a small deletion in exon 28 in a BM patient [22]. This case highlighted the relevance of array-CGH as a useful complementary diagnostic tool, especially in recessive forms of the disease in which only one mutant allele is detected by standard sequencing.

Unlike published patient cohorts [16,41,42], milder phenotypes (BM and INT/BM) were prevalent in our series, accounting for 59% of the patients. The recognition of mild phenotypes is certainly enhanced by the activity within clinical networks of excellence that bring together highly specialized centers, such as the European Reference Network (ERN) for neuromuscular diseases (ERN EURO-NMD) [43].

The ability to identify milder phenotypes and accurately categorize patients with collagen VI-related myopathies into phenotype subgroups will be crucial in optimizing clinical care, in the design of clinical trials for the therapeutic strategies currently under development, and in the definition of natural history and cohort studies.

Although meaningful genotype–phenotype correlations were not identified in our study, variants in *COL6A1* tended to be associated with more severe phenotypes (UCMD), whereas less severe phenotypes (BM) were observed in the *COL6A2* and *COL6A3* cohorts.

The majority of patients presented with a monoallelic *COL6A* gene variant, suggesting an autosomal dominant/de novo inheritance (78.3%). Biallelic variants were found in 21.7% of the patients.

Notably, patients carrying dominantly acting variants appear to be more common in several other studies [7,30,41,44]. Rare exceptions are the small French cohorts, in which the number of dominant and recessive cases did not differ significantly [42], and the Egyptian cohort, in which patients with recessive variants predominated (56.5%), probably due to the higher inbreeding rates in this population [17].

In our cohort, biallelic variants in *COL6A1* invariably led to a UCMD phenotype, whereas all phenotypes were associated with biallelic *COL6A2* and *COL6A3* variants. Unlike UCMD, which notoriously displays both autosomal dominant and recessive inheritance, BM has been considered exclusively an autosomal dominant disease for a long time. However, in recent years, few cases of recessive inheritance were described [21,45,46,47,48,49]. Our work significantly contributes to the collection of “milder” autosomal recessive cases because 10 of our BM patients presented a recessive inheritance. Intriguingly, nine of these (90%) displayed biallelic variants in the *COL6A2* gene, mostly concentrated in the C-terminal end of the α2(VI) chain.

This finding might indicate that biallelic variants in the *COL6A2* gene are more likely to be tolerated than biallelic variants in the other *COL6A* genes, thus leading to milder phenotypes. However, this hypothesis deserves to be explored with further extensive patient studies.

Concerning the involved protein domain, variants in the *COL6A1* and *COL6A3* genes were almost exclusively located in the NH2 and THD regions, consistent with data from other works [7,16,31].

In contrast, in the *COL6A2* gene, variants were distributed throughout the gene with a prevalence in the carboxy-terminal domain. Interestingly, this finding is consistent with data from a large Chinese multicenter study in which the mutations were observed along the entire length of the *COL6A2* gene, from the N-terminus to the THD and C-terminus [33].

On the tetramer structure of collagen VI, N-terminal globular domains are believed to be an attaching site for microfibril formation, but one hypothesis is that the carboxy-terminal domain is also crucial for this [50]. Indeed, after synthesis, the three collagen VI chains associate through the C-terminal globular domain and coil into a triple-helical monomer. The monomers are then assembled in a staggered, antiparallel manner into dimers, which then associate laterally into tetramers. The tetramers are secreted outside of the cells, where they assemble into collagen VI microfibrils [50].

A specific role for the C-terminal globular domain has been proposed by Zhang and colleagues, who suggested that this domain is critical for the proper alignment of the tetramer and for heterotypic interactions with other matrix molecules [40].

In addition, recessive mutations in the carboxy-terminal domain of the *COL6A2* gene have been reported in the literature associated with a severe UCMD phenotype, emphasizing the importance of this region [50,51].

The *COL6A2* mutation spectrum (all mutation types), together with its capability to be mutated in all collagen VI phenotypes, is interesting. Collagen gene preservation was checked in several species, from fungi to primates, and while *COL6A1* and *COL6A3* were not present in chimpanzees and rats, *COL6A2* was found to be preserved in species ranging from fish to primates [52]. More generally, the *COL6A* genes were found to be the most ancient of all collagen-encoding genes. This peculiar evolutive aspect might be related to the unique mutation scenario and phenotypic association of *COL6A2*.

## 5. Conclusions

To the best of our knowledge, this paper describes the largest cohort of genotyped patients worldwide diagnosed in a temporal window of 15 years in 21 Italian diagnostic centers.

Our data on this large cohort of collagen VI-related myopathy patients reinforces previous reports describing the variability of the clinical spectrum, the considerable allelic heterogeneity and the resulting complexity of diagnosis [53].

Recent innovations in molecular genetics and innovative approaches [54], including the use of artificial intelligence (AI) [55], have provided tremendous support in the diagnostic process for conditions characterized by pronounced heterogeneity, such as collagen VI-related myopathies.

Also extremely significant in this context is the advantageous networking strategy offered by the ERN EURO-NMD [43]. Sharing expertise through ERNs has greatly improved diagnostic accuracy, the ability to identify the milder phenotypes and the categorization of patients into subgroups. Thanks to this network, Health Care Providers (HCPs) can easily interface with each other and use dedicated IT tools, such as the Clinical Patient Management System (CPMS), an innovative digital platform for discussing complex and atypical clinical cases [56].

An important and novel finding that has emerged from our study is how the *COL6A2* gene has peculiar characteristics that distinguish it from the *COL6A1* and *COL6A3* genes. First, the *COL6A2* gene was found to be the most frequently involved, with the greatest allelic variability and the full spectrum of variant types. These variants are both dominant and recessive, the latter being much more common than in the other *COL6A* genes. The distribution of variants along the α2(VI) chain is also characteristic, not limited to the N-terminal end of the THD, but also frequently occurring in the C-terminal domain, which appears to be of primary importance in the complex process of protein assembly. Although the small number of patients reported to date hampers definitive conclusions, it is interesting that *COL6A2* is the only gene affected in our myosclerosis myopathy patients, suggesting that this should be the first gene to be investigated when the phenotype is highly suggestive.

This diversity is surprising, considering that *COL6A1* and *COL6A2* genes have a highly similar sequence and protein structure, cluster on the same 21q22.3 chromosomal region and share an ancestral gene duplication origin. However, evolutionary and structural differences between the two genes exist, as the *COL6A2* gene is more ancient than the *COL6A1* gene in animal phylogeny, is the unique *COL6A* gene that is preserved in all vertebrates without gene losses [52]. Additionally, and significantly, the human intron–exon organization of the globular domains of *COL6A2* and *COL6A1* is different, in contrast with the conservation of the triple-helical exon organization [57]. These differences could be the expression of different functions and deserve further investigations to understand their correlation with the *COL6A2* mutation pattern and their implications in the pathophysiology of collagen VI-related myopathies.

## Figures and Tables

**Figure 1 biomolecules-15-01426-f001:**
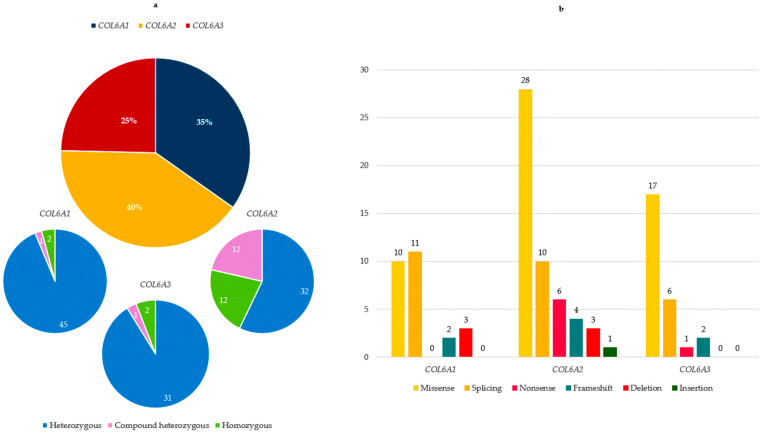
(**a**) *COL6A* gene distribution among patients of our cohort. For each *COL6A* gene, proportion of heterozygous, compound heterozygous and homozygous is reported; (**b**) type and number of variants in the three *COL6A* genes.

**Figure 2 biomolecules-15-01426-f002:**
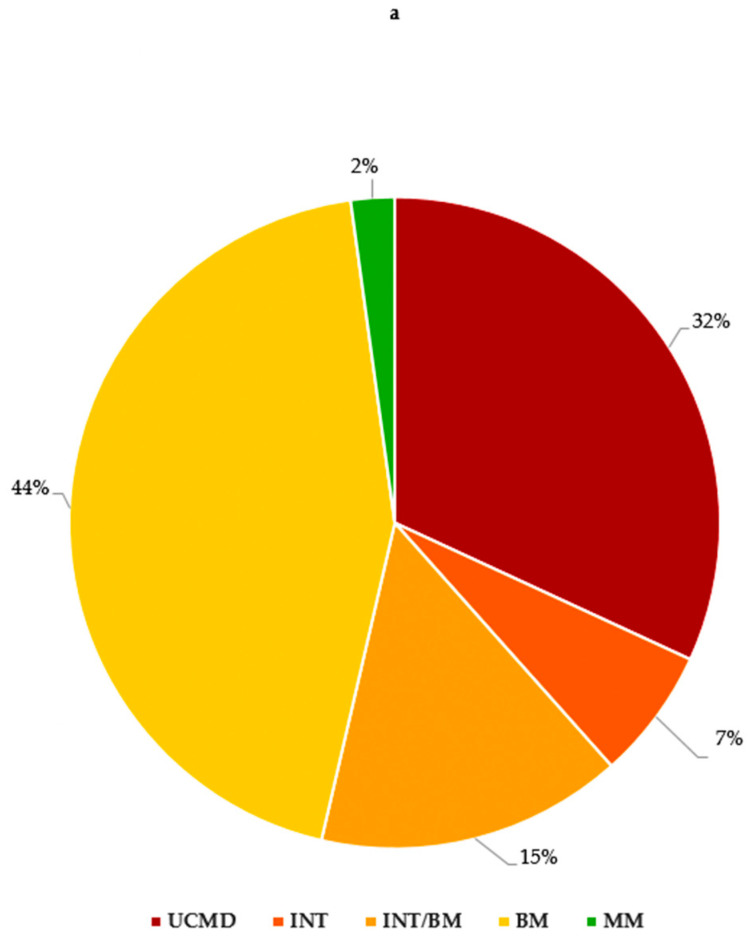
(**a**) *COL6A* gene and phenotypes observed in our cohort; (**b**) number of variants in *COL6A* genes for each phenotype. UCMD: Ullrich congenital muscular dystrophy; INT: intermediate collagen VI-related myopathy; INT/BM: intermediate collagen VI-related myopathy/Bethlem myopathy; BM: Bethlem myopathy; MM: myosclerosis myopathy.

**Figure 3 biomolecules-15-01426-f003:**
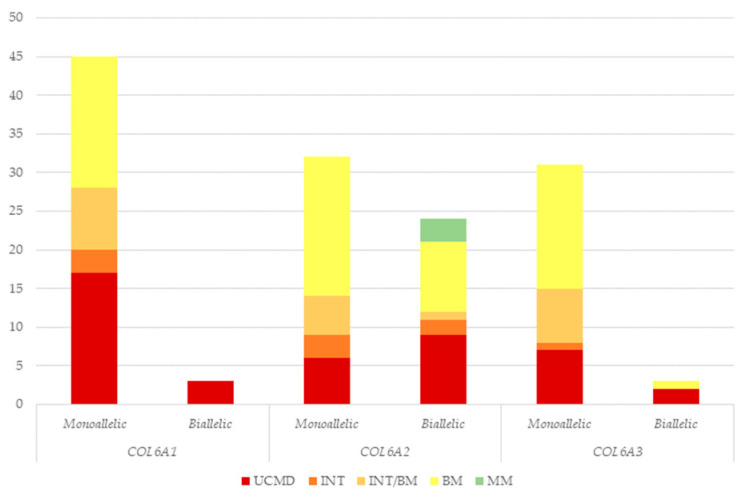
Number of patients presenting monoallelic or biallelic *COL6A* gene variants according to each phenotype. UCMD: Ullrich congenital muscular dystrophy; INT: intermediate collagen VI-related myopathy; INT/BM: intermediate collagen VI-related myopathy/Bethlem myopathy; BM: Bethlem myopathy; MM: myosclerosis myopathy.

**Figure 4 biomolecules-15-01426-f004:**
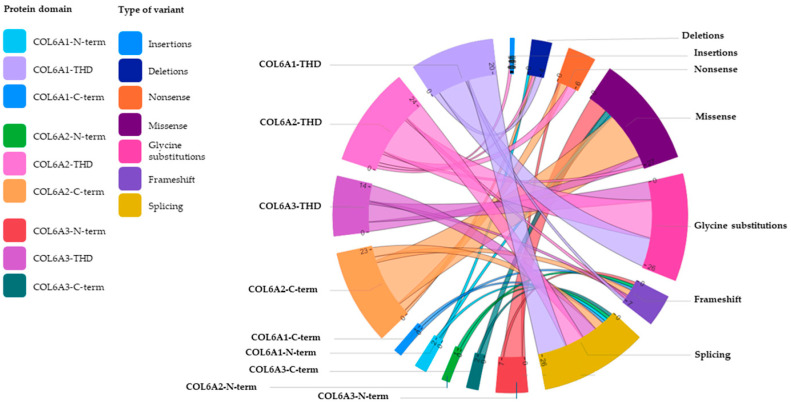
Schematic representation of the mutation distribution within different domains of the three *COL6A* genes. The innovative diagram, called Chord (Microsoft Corporation), enables us to intuitively visualize the relationships between several variables, in this case between the three domains (N-terminal, TH and C-terminal) of the *COL6A1*, *COL6A2* and *COL6A3* genes and the different types of variants. Each variable, assigned a color code, is placed at a precise point, defined as a ‘node’, along a circular layout and connected to other nodes by arcs (literally ‘strings’). Each connection is assigned a value (in this case, corresponding to the number of variants), which is represented proportionally by the size of each arc, with thicker strings indicating numerically more significant connections and thinner strings representing weaker connections. THD: triple helical domain; N-term: N-terminal; C-term: C-terminal.

**Table 1 biomolecules-15-01426-t001:** Clinical and genetic data of patients of the *COL6A1*, *COL6A2* and *COL6A3* cohorts.

*COL6A1 (NM_001848.2)*													
ID	M / F	MA / BA	Genotype	Exon / Intron	cDNA Change	Protein Change	Domain	Mutation Type	In Silico Prediction	Age at Last Evaluation (y)	Walking Ability Acquired	Still Walking	Age at Loss of Deambulation (y)
** UCMD **													
**P1**	F	MA	het	exon 8-intron 8	c.798_804+8del	p.Pro254_Glu268del	N-THD	deletion	LP	20	yes	no	6
**P2**	M	MA	het	exon 9	c.819_833del	p. Pro274_Gly278del	THD	deletion	LP	11	no	_	_
**P3**	M	MA	het	exon 9	c.850G>A	p.Gly284Arg	THD	missense	P	10	yes	no	10
**P4**	F	MA	het	exon 9	c.850G>A	p.Gly284Arg	THD	missense	P	10	no	_	_
**P5**	F	MA	het	exon 9	c.850G>A	p.Gly284Arg	THD	missense	P	26	yes	no	9
**P6**	M	MA	het	exon 9	c.850G>A	p.Gly284Arg	THD	missense	P	10	yes	no	12
**P7**	F	MA	het	exon 9	c.850G>A	p.Gly284Arg	THD	missense	P	3	yes	yes	_
**P8**	F	MA	het	exon 9	c.850G>A	p.Gly284Arg	THD	missense	P	6.5	yes	yes	_
**P9**	M	MA	het	exon 10	c.868G>A	p.Gly290Arg	THD	missense	P	2	no	_	_
**P10**	M	MA	het	exon 10	c.877G>A	p.Gly293Arg	THD	missense	P	8	yes	no	8
**P11**	F	MA	het	exon 10	c.893_898del	p.Gln298_Met300delinsLeu	THD	deletion	LP	3	no	_	_
**P12**	F	MA	het	intron 10	c.904-2A>G	p.Gly302_Lys310del	THD	splicing	P	3	no	_	_
**P13**	M	MA	het	intron 10	c.904-10G>A	p.Gly302_Lys310del	THD	splicing	VUS	15	yes	no	9
**P14**	F	MA	het	intron 11	c.930+189C>T	p.?	THD	splicing	LP	10	yes	no	6.5
**P15**	F	MA	het	intron 11	c.930+189C>T	p.?	THD	splicing	LP	13	yes	no	12
**P16**	F	MA	het	intron 11	c.930+189C>T	p.?	THD	splicing	LP	8.5	yes	no	4.5
**P17**	F	MA	het	intron 11	c.930+189C>T	p.?	THD	splicing	LP	11	yes	no	9
**P18**	M	BA	homo	exon 22	c.1465del	p.Ala489ProfsTer16	THD	frameshift	P	10	no	_	_
**P19**	M	BA	homo	exon 24	c.1576G>A	p.Gly526Arg	THD	missense	VUS	12	yes	no	6
**P20**	M	BA	comp. het	intron 32	c.2250+1G>C	p.?		splicing	P	9	yes	no	5
				exon 33	c.2327_2330dup	p.Ala778ProfsTer52	C (vWFA2)	frameshift	LP				
** INT **													
**P21**	M	MA	het	exon 9	c.841G>A	p.Gly281Arg	THD	missense	P	33	yes	no	17
**P22**	F	MA	het	exon 10	c.877G>A	p.Gly293Arg	THD	missense	P	45	yes	no	18
**P23**	F	MA	het	exon 10	c.896G>A	p.Gly299Glu	THD	missense	P	21	yes	no	19
** INT/BM **													
**P24**	F	MA	het	exon 9	c.850G>A	p.Gly284Arg	THD	missense	P	11	yes	yes	_
**P25**	F	MA	het	exon9	c.851G>A	p.Gly284Glu	THD	missense	LP	10	yes	yes	_
**P26**	M	MA	het	exon 10	c.877G>A	p.Gly293Arg	THD	missense	P	15	yes	yes	_
**P27**	F	MA	het	exon 10	c.877G>A	p.Gly293Arg	THD	missense	P	11	yes	yes	_
**P28**	F	MA	het	intron 13	c.1003-1G>A	p.?	THD	splicing	P	14.5	yes	yes	_
**P29**	F	MA	het	intron 14	c.1056+1G>A	p.Gly335_Asp352del	THD	splicing	P	9	yes	yes	_
**P30**	M	MA	het	intron 14	c.1056+1G>C	p.Gly335_Asp352del	THD	splicing	P	12	yes	yes	_
**P31**	M	MA	het	intron 14	c.1056+5G>A	p.Gly335_Asp352del	THD	splicing	LP	23	yes	yes	_
** BM **													
**P32**	F	MA	het	exon 3	c.362A>G	p.Lys121Arg	N (vWFA1)	missense	LP	30	yes	yes	_
**P33**	F	MA	het	intron 3	c.428+1G>A	p.Tyr122_Gly143del	N (vWFA1)	splicing	P	28	yes	yes	_
**P34**	F	MA	het	exon 9	c.850G>A	p.Gly284Arg	THD	missense	P	20	yes	yes	_
**P35**	M	MA	het	exon 10	c.868G>C	p.Gly290Arg	THD	missense	P	35	yes	yes	_
**P36**	F	MA	het	exon 10	c.868G>A	p.Gly290Arg	THD	missense	P	37	yes	yes	_
**P37**	M	MA	het	exon 10	c.877G>A	p.Gly293Arg	THD	missense	P	36	yes	yes	_
**P38**	F	MA	het	exon 10	c.877G>A	p.Gly293Arg	THD	missense	P	40	yes	yes	_
**P39**	F	MA	het	intron 11	c.930+1 G>A	p.?	THD	splicing	P	39	yes	yes	_
**P40**	M	MA	het	intron 11	c.930+1 G>A	p.?	THD	splicing	P	44	yes	yes	_
**P41**	F	MA	het	intron 11	c.930+1G>A	p.?	THD	splicing	P	42	yes	yes	_
**P42**	M	MA	het	intron 11	c.931-3C>G	p.?	THD	splicing	VUS	22	yes	yes	_
**P43**	F	MA	het	exon 14	c.1022G>A	p.Gly341Asp	THD	missense	P	54	yes	yes	_
**P44**	F	MA	het	intron 14	c.1056+1G>A	p.?	THD	splicing	P	45	yes	no	43
**P45**	F	MA	het	intron 14	c.1056+1G>A	p.Gly335_Asp352del	THD	splicing	P	33	yes	yes	_
**P46**	M	MA	het	intron 14	c.1056+1G>A	p.Gly335_Asp352del	THD	splicing	P	21	yes	yes	_
**P47**	F	MA	het	intron 14	c.1056+1G>A	p.Gly335_Asp352del	THD	splicing	P	27	yes	yes	_
**P48**	F	MA	het	intron14	c.1056+1G>A	p.Gly335_Asp352del	THD	splicing	P	30	yes	yes	_
** *COL6A2 (NM_001849.3)* **													
**ID**	**M** **/** **F**	**MA/** **BA**	**Genotype**	**Exon** **/** **Intron**	**cDNA Change**	**Protein Change**	**Domain**	**Mutation** **Type**	**In Silico Prediction**	**Age at Last Evaluation (y)**	**Walking Ability Acquired**	**Still Walking**	**Age at Loss of Deambulation (y)**
** UCMD **													
** P49 **	F	BA	homo	exon 3	c.348dup	p.Ser117LeufsTer159	N (vWFA1)	frameshift	LP	19	yes	no	12
** P50 **	F	BA	comp. het	exon 3	c.348dup	p.Ser117LeufsTer159	N (vWFA1)	frameshift	LP	31	no	_	_
				exon 15	c.1327G>T	p.Glu443Ter	THD	nonsense	P				
** P51 **	F	MA	het	exon 7	c.875G>T	p.Gly292Val	THD	missense	P	29	yes	no	6
** P52 **	F	MA	het	exon 9	c.954G>A	p.Lys318Lys	THD	missense	LP	24	yes	no	10
** P53 **	M	MA	het	intron 9	c.954+23_955-43del	p.Gly310_Lys318del	THD	deletion	VUS	19	yes	no	6
** P54 **	F	MA	het	int 15/exon 16	c.1333-1/1336insG	p.Asp446GlyfsTer5	THD	frameshift	LP	8	no	_	_
** P55 **	F	BA	homo	exon 17	c.1402C>T	p.Arg468Ter	THD	nonsense	P	2	yes	yes	_
** P56 **	M	BA	comp. het	intron 17	c.1459-2A>G	p.Gly487AspfsTer48	THD	splicing	P	12	yes	no	8
				intron 23	c.1771-1G>A	p.Glu591ThrfsTer148	C	splicing	P				
** P57 **	F	BA	comp. het	exon 21	c.1619G>A	p.Gly540Asp	THD	missense	LP	18	no	_	_
				exon 26	c.2329T>C	p.Cys777Arg	C (vWFA2)	missense	LP				
** P58 **	M	BA	comp. het	exon 26	c.2098G>A	p.Gly700Ser	C (vWFA2)	missense	P	6.5	yes	no	8
				exon 26	c.2381C>A	p.Ala794Asp	C (vWFA2)	missense	VUS				
** P59 **	F	BA	homo	exon 26	c.2175_2176del	p.Phe726CysfsTer15	C (vWFA2)	frameshift	LP	13	yes	no	7
** P60 **	M	MA	het	exon 27	c.2455C>T	p.Gln819Ter	C	nonsense	P	7	yes	no	6
** P61 **	F	MA	het	exon 28	c.2560C>T	p.Arg854Cys	C (vWFA3)	missense	VUS	3	yes	yes	_
** P62 **	M	BA	homo	exon 28	c.2572C>T	p.Gln858Ter	C (vWFA3)	nonsense	P	8	no	_	_
** P63 **	M	BA	homo	exon 28	c.2626C>A	p.Arg876Ser	C (vWFA3)	missense	LP	9	yes	no	6
** INT **													
** P64 **	M	MA	het	intron 5	c.801+3A>C	p.Cys246_Lys267del	N-THD	splicing	VUS	15	yes	no	15
** P65 **	F	MA	het	intron 6	c.855+1G>A	p.?		splicing	P	17	yes	no	14
** P66 **	M	MA	het	exon 7	c.875G>A	p.Gly292Asp	THD	missense	P	21	yes	no	14
** P67 **	M	BA	comp. het	intron 8	c.927+5G>A	p.Lys318fsTer6	THD	splicing	LP	17	yes	no	17
				exon 12	c.1096C>T	p.Arg366Ter	THD	nonsense	P				
** P68 **	F	BA	homo	exon 28	c.2572C>T	p.Gln858Ter	C (vWFA3)	nonsense	P	22	yes	no	15
** INT/BM **													
** P69 **	F	MA	het	intron 5	c.802-2A>T	p.?		splicing	LP	14	yes	yes	_
** P70 **	F	MA	het	exon 6	c.847G>A	p.Gly283Arg	THD	missense	P	16.5	yes	yes	_
** P71 **	F	MA	het	exon 7	c.884G>T	p.Gly295Val	THD	missense	P	11	yes	yes	_
** P72 **	M	MA	het	exon 7	c.893G>A	p.Gly298Glu	THD	missense	P	17	yes	yes	_
** P73 **	F	BA	homo	intron 25	c.1970-9G>A	p.Thr656fsTer18	C (vWFA2)	splicing	P	17	yes	yes	_
** P74 **	M	MA	het	exon 26	c.2060T>C	p.Phe687Ser	C (vWFA2)	missense	VUS	17	yes	yes	_
** BM **													
** P75 **	F	MA	het	exon 6	c.802G>A	p.Gly268Ser	THD	missense	P	58	yes	no	56
** P76 **	M	MA	het	exon 6	c.802G>A	p.Gly268Ser	THD	missense	P	32	yes	yes	_
** P77 **	F	MA	het	exon 6	c.830G>A	p.Gly277Glu	THD	missense	LP	20	yes	yes	_
** P78 **	M	MA	het	exon 6	c.847G>A	p.Gly283Arg	THD	missense	P	40	yes	no	31
** P79 **	F	MA	het	exon 7	c.875G>C	p.Gly292Ala	THD	missense	LP	64	yes	no	51
** P80 **	F	MA	het	exon 7	c.883G>A	p.Gly295Arg	THD	missense	P	25	yes	yes	_
** P81 **	M	MA	het	exon 8	c.902G>A	p.Gly301Asp	THD	missense	P	28	yes	yes	_
** P82 **	F	MA	het	exon 8	c.911G>T	p.Gly304Val	THD	missense	LP	2	yes	yes	_
** P83 **	M	MA	het	intron 13	c.1179+2T>G	p.?	THD	splicing	LP	53	yes	yes	_
** P84 **	M	MA	het	intron 23	c.1770+1G>A	p.Gly579_Thr590del	THD	splicing	LP	41	yes	no	40
** P85 **	F	BA	comp. het	exon 25	c.1832G>A	p.Cys611Tyr	C	missense	VUS	36	yes	yes	_
				exon 26	c.2329T>C	p.Cys777Arg	C (vWFA2)	missense	LP				
** P86 **	F	MA	het	exon 25	c.1861G>A	p.Asp621Asn	C (vWFA2)	missense	P	31	yes	yes	_
** P87 **	M	MA	het	exon 25	c.1867T>G	p.Ser623Ala	C (vWFA2)	missense	LP	56	yes	yes	_
** P88 **	M	MA	het	intron 25	c.1970-3C>A	p.Thr656_Ala698del	C (vWFA2)	splicing	VUS	37	yes	yes	_
** P89 **	M	BA	homo	intron 25	c.1970-9G>A	p.Thr656fsTer18	C (vWFA2)	splicing	P	20	yes	yes	_
** P90 **	F	BA	comp. het	intron 25	c.1970-9G>A	p.Thr656fsTer18	C (vWFA2)	splicing	P	22	yes	yes	_
				exon 26	c.1970-7_1981dup	p.Gly661AlafsTer18	C (vWFA2)	frameshift	LP				
** P91 **	F	BA	comp. het	intron 25	c.1970-9G>A	p.Thr656fsTer18	C (vWFA2)	splicing	P	25	yes	yes	_
				exon 28	c.2489G>A	p.Arg830Gln	C	missense	LP				
				exon 28	c.2527C>T	p.Arg843Trp	C (vWFA3)	missense	VUS				
** P92 **	M	BA	homo	exon 26	c.2060T>C	p.Phe687Ser	C (vWFA2)	missense	VUS	51	yes	yes	_
** P93 **	M	MA	het	exon 26	c.2098G>A	p.Gly700Ser	C (vWFA2)	missense	P	38	yes	yes	_
** P94 **	M	MA	het	exon 26	c.2192C>T	p.Thr731Met	C (vWFA2)	missense	P	56	yes	yes	_
** P95 **	M	MA	het	exon 26	c.2192C>T	p.Thr731Met	C (vWFA2)	missense	P	50	yes	yes	_
** P96 **	M	MA	het	exon 26	c.2192C>T	p.Thr731Met	C (vWFA2)	missense	P	46	yes	yes	_
** P97 **	M	BA	homo	exon 26	c.2240T>A	p.Leu747Gln	C (vWFA2)	missense	VUS	51	yes	no	45
** P98 **	M	BA	homo	exon 27	c.2455C>T	p.Gln819Ter	C	nonsense	P	55	yes	no	48
** P99 **	M	MA	het	exon 28	c.2528G>A	p.Arg843Gln	C (vWFA3)	missense	VUS	38	yes	yes	_
** P100 **	F	BA	comp. het	exon 28	c.2947_2952del	p.Asp983_Val984del	C (vWFA3)	deletion	VUS	30	yes	yes	_
				intron 1	chr21 g(46352739-46352798)_(46354936)del Genome Build NCBI35/hg17 *	p.?		deletion	VUS				
** P101 **	F	BA	comp. het	exon 28	c.2665C>T	p.Gln889Ter	C (vWFA3)	nonsense	LP	24	yes	yes	_
				exon 5	c.780_781insCCCCCC	p.Pro260_Lys261insProPro	THD	insertion	LP				
** MM **													
**P102**	M	BA	comp. het	exon 12	c.1096C>T	p.Arg366Ter	THD	nonsense	P	49	yes	no	48
				exon 28	c.2611G>A	p.Asp871Asn	C (vWFA3)	missense	LP				
**P103**	F	BA	homo	exon 27	c.2455C>T	p.Gln819Ter	C	nonsense	P	12	yes	no	11
**P104**	F	BA	comp. het	exon 27	c.2455C>T	p.Gln819Ter	C	nonsense	P	32	yes	yes	_
				exon 28	c.2489G>A	p.Arg830Gln	C	missense	LP				
** *COL6A3 (NM_004369.3)* **													
**ID**	**M** **/** **F**	**MA/** **BA**	**Genotype**	**Exon** **/** **Intron**	**cDNA Change**	**Protein Change**	**Domain**	**Mutation** **Type**	**In Silico Prediction**	**Age at Last Evaluation (y)**	**Walking Ability Acquired**	**Still Walking**	**Age at Loss of Deambulation (y)**
** UCMD **													
** P105 **	M	MA	het	intron 16	c.6210+1G>A	p.Gly2053_Pro2070del	THD	splicing	P	23	no	_	_
** P106 **	F	MA	het	intron 16	c.6210+1G>A	p.Gly2053_Pro2070del	THD	splicing	P	13	yes	no	12.5
** P107 **	M	MA	het	intron 16	c.6210+1G>A	p.Gly2053_Pro2070del	THD	splicing	P	26	no	_	_
** P108 **	M	MA	het	intron 16	c.6210+5G>A	p.?	THD	splicing	VUS	13	yes	no	11
** P109 **	F	MA	het	intron 17	c.6282+1G>A	p.?	THD	splicing	P	18	yes	no	6
** P110 **	F	MA	het	intron 17	c.6282+1G>A	p.?	THD	splicing	P	29	yes	no	9
** P111 **	M	BA	homo	exon 18	c.6284del	p.Gly2095AlafsTer12	THD	frameshift	LP	10	yes	no	7
** P112 **	M	MA	het	exon 28	c.6871G>C	p.Gly2291Arg	THD	missense	LP	8.5	yes	no	6
** P113 **	M	BA	comp. het	exon 32	c.7066G>A	p.Gly2356Arg	THD	missense	LP	11	no	_	_
				intron 36	c.7669-3C>G	p.?		splicing	VUS				
** INT **													
** P114 **	M	MA	het	intron 16	c.6210+1G>A	p.Gly2053_Pro2070del	THD	splicing	P	25	yes	no	17
** INT/BM **													
** P115 **	F	MA	het	exon 7	c.2536G>A	p.Ala846Thr	N (vWFA5)	missense	VUS	12	yes	yes	_
** P116 **	M	MA	het	exon 9	c.4121A>T	p.Asp1374Val	N (vWFA7)	missense	VUS	17	yes	yes	_
** P117 **	F	MA	het	exon 12	c.5794_5797dup	p.Tyr1933CysfsTer22	N (vWFA10)	frameshift	LP	14	yes	yes	_
** P118 **	M	MA	het	intron 15	c.6156+2T>G	p.?		splicing	LP	15	yes	yes	_
** P119 **	M	MA	het	exon 16	c.6199G>A	p.Glu2067Lys	THD	missense	LP	18	yes	yes	_
** P120 **	M	MA	het	exon 17	c.6229G>C	p.GLy2077Arg	THD	missense	LP	10	yes	yes	_
** P121 **	F	MA	het	exon 36	c.7254C>A	p.Phe2418Leu	C (vWFA11)	missense	VUS	10	yes	yes	_
** BM **													
** P122 **	F	BA	homo	exon 5	c.1393C>T	p.Arg465Ter	N (vWFA3)	nonsense	P	35	yes	yes	_
** P123 **	M	MA	het	exon 10	c.4859C>T	p.Pro1620Leu	N	missense	VUS	23	yes	yes	_
** P124 **	M	MA	het	exon 11	c.4928T>G	p.Leu1643Arg	N (vWFA9)	missense	LP	19	yes	yes	_
** P125 **	M	MA	het	exon 11	c.5035G>T	p.Gly1679Trp	N (vWFA9)	missense	LP	55	yes	yes	__
** P126 **	M	MA	het	exon 11	c.5035G>T	p.Gly1679Trp	N (vWFA9)	missense	LP	26	yes	yes	_
** P127 **	M	MA	het	exon 11	c.5035G>T	p.Gly1679Trp	N (vWFA9)	missense	LP	33	yes	yes	_
** P128 **	M	MA	het	exon 12	c.5524G>A	p.Gly1842Arg	N (vWFA10)	missense	LP	57	yes	yes	_
** P129 **	M	MA	het	exon 12	c.5794_5797dup	p.Tyr1933CysfsTer22	N (vWFA10)	frameshift	LP	43	yes	yes	_
** P130 **	M	MA	het	exon 15	c.6156G>T	p.Lys2052Asn	THD	missense	P	22	yes	yes	_
** P131 **	F	MA	het	exon 16	c.6158G>T	p.Gly2053Val	THD	missense	P	35	yes	yes	_
** P132 **	M	MA	het	exon 16	c.6158G>T	p.Gly2053Val	THD	missense	P	20	yes	yes	_
** P133 **	F	MA	het	exon 16	c.6166G>A	p.Gly2056Arg	THD	missense	LP	41	yes	yes	_
** P134 **	M	MA	het	exon 16	c.6199G>A	p.Glu2067Lys	THD	missense	LP	19	yes	yes	_
** P135 **	F	MA	het	intron 16	c.6210+1G>C	p.Gly2053_Pro2070del	THD	splicing	LP	19	yes	yes	_
** P136 **	F	MA	het	exon 17	c.6230G>A	p.Gly2077Asp	THD	missense	P	42	yes	yes	_
** P137 **	F	MA	het	exon 28	c.6820G>A	p.Glu2274Lys	THD	missense	VUS	26	yes	yes	_
** P138 **	M	MA	het	exon 36	c.7468G>A	p.Ala2490Thr	C (vWFA11)	missense	VUS	20	yes	yes	_

M: Male; F: Female; MA: Monoallelic; BA: Biallelic; Het: heterozygous; Homo: homozygous; Comp het: compound heterozygous; UCMD: Ullrich congenital muscular dystrophy; INT: intermediate collagen VI-related myopathy; INT/BM: intermediate collagen VI-related myopathy/Bethlem myopathy; BM: Bethlem myopathy; MM: myosclerosis myopathy: P: pathogenic; LP: likely pathogenic; VUS: variant of uncertain significance. * GRCh37/hg19: g.47528311_47528370)_(47530470_47530508).

## Data Availability

All variants presented in this study have been submitted to the LOVD gene-specific database for free consultation.

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
