# Peer review of "Landscape Analysis of *COL6A1*, *COL6A2*, and *COL6A3* Pathogenic Variants in a Large Italian Cohort Presenting with Collagen VI-Related Myopathies: A Nationwide Report"

_biomolecules, 2025, doi:10.3390/biom15101426_

Round 1

Reviewer 1 Report

Comments and Suggestions for Authors

The manuscript is well  written and presents data that should be relevant for the scientific community and especially in regards to clinical handling of the patient group with Collagen VI-related myopathies.

In the introduction presenting the genes and associated disorders a description of usual modes of inheritance will be appreciated.

Under genetic analysis the term "coverage" is used both to describe read depth and the proportion of the targeted region that has been sequenced. I recomemend using "read depth" and "coverage", respectively (https://3billion.io/blog/sequencing-depth-vs-coverage)

How are alignment and variant calling performed?

The sentence line 139, indicates that databases with a frequency of 0.1% were filtered. Reformulate.

The ABIPRISM is not an automated analyzer, but either a sequence analyzer or a genetic analyzer

In table one NM number of the transcripts should be mentioned. A large number of the variants has previously been described refrences should be included.

For a number of splice variants shown results are indicated as either frameshifts or terminations. How is this shown. Did you perform protein sequencing? In the original refrences the consequence is only predicted - thus the nomenclature may be inappropriate.

Was haplotyping performed for indiuviduals with identical genotypes to establish if these are individual founders, hence representing hot spots, or actually distent relatives.

The quality of figure 1 should be improved to increase redability. In the legend phenotypes are indicaated but these are not addressed in the figure. Same for figure 2.

In line 201 the authors mention "de novo inheritance" did tou investigate for mosaicisme or uneven VAF which could indicate this?

In the same line "multiple" indicates "many" i.e. more than two. 

For discussion; what are the mechanisms when both nonsense and missense variants can be dominant negative. 

The hyphens in line 75 and 76 seems misplaced

Reviewer 2 Report

Comments and Suggestions for Authors

The paper presents the results of a clinical and molecular genetic examination of patients with variants in the genes of type 6 collagens with the muscular dystrophy phenotype.

Major remark:

Monoallelic (heterozygous variants) were found in 78% of patients. The authors claim that "The majority of patients presented a monoallelic COL6A gene variant, suggesting an autosomal dominant/de novo 316 inheritance (78.3%)." (lines 316-317). But the authors also claim that no family analysis was performed. The question arises as to the status of these heterozygous variants in terms of pathogenicity/causality. Diseases associated with type 6 collagen genes can be inherited by both AD and AR types, so the causality of heterozygous variants needs to be discussed.

Minor remark:

The authors should note the variants of the nucleotide sequence previously described and identified for the first time in this work. For those newly identified, a more detailed discussion of the phenotype and pathogenicity is desirable.

Reviewer 3 Report

Comments and Suggestions for Authors

The manuscript by Fortunato et al. presents the results of a nationwide analysis of the COL6A1, COL6A2, and COL6A3 genes to search for pathogenic variants associated with collagen VI-related myopathies. This study is one of the largest in this field, but there are some issues that need to be addressed before acceptance.

  1. The manuscript reports that 138 Italian patients were analyzed, and 104 variants were identified among them. However, the manuscript does not clarify how many patients were suspected of having collagen VI-related myopathy but did not have any mutations in the COL6A1, COL6A2, or COL6A3 genes. It would be helpful to include this information.
  2. Additionally, it would be beneficial to provide a diagram or flowchart that outlines the diagnostic algorithm for diagnosing collagen VI-related myopathies based on the results of the genetic analysis.

Reviewer 4 Report

Comments and Suggestions for Authors

See attached.

Comments on the Quality of English Language

Generally good. Some phrases can be more succinctly expressed. Some were recommended in the attachment.

Round 2

Reviewer 3 Report

Comments and Suggestions for Authors

In its current form, the manuscript reads like a report on molecular genetic diagnostics accumulated over many years. While the volume of diagnostic data is impressive, the overall study falls outside the scope of this journal. I recommend transferring this manuscript to a medical genetics journal. 

Author Response

We are very surprised and disappointed by this comment. Our article is based on a personal invitation to Prof Ferlini made on May 12th, she received from the Journal (Mrs Florence Liu). We therefore proposed the publication of a large case series of patients with COL6-related myopathies and submitted the abstract of the work. Our proposal has got a fully positive editorial evaluation, and it was suggested that, if accepted, the manuscript could be included in the Special Issue: “State-of-the-Art of Myology 2024–2025”. Based on these agreements, we submitted the manuscript. Therefore, we do not have any answer for this referee’ comment.

Reviewer 4 Report

Comments and Suggestions for Authors

Thanks for addressing comments from my initial review.

Author Response

We would like to thank the reviewer for the attention and care devoted to our work.